# The baseline immunological and hygienic status of pigs impact disease severity of African swine fever

Emilia Radulovic[1,2,3☺], Kemal Mehinagic[1,2,3☺], Tsering Wüthrich[3,4], Markus Hilty[4], Horst Posthaus[2], Artur Summerfield[1,2,5‡], Nicolas Ruggli[1,2‡*], Charaf Benarafa[1,2,5‡*]

**1** Institute of Virology and Immunology IVI, Mittelhäusern, Switzerland, **2** Vetsuisse Faculty, Department of Infectious Diseases and Pathobiology, University of Bern, Bern, Switzerland, **3** Graduate School for Cellular and Biomedical Science, University of Bern, Bern, Switzerland, **4** Institute for Infectious Diseases, University of Bern, Bern, Switzerland, **5** Multidisciplinary Center for Infectious Diseases, University of Bern, Bern, Switzerland

☺ These authors contributed equally to this work.
‡ These authors co-supervised the study.
* Nicolas.Ruggli@ivi.admin.ch (NR); charaf.benarafa@vetsuisse.unibe.ch (CB)

**Data Availability Statement:** The authors confirm that all data underlying the findings are fully available without restriction and were uploaded on Zenodo 10.5281/zenodo.6831560. All sequencing

## Abstract

African Swine Fever virus (ASFV) is a large double-enveloped DNA virus of the *Asfarviridae* family that causes a lethal hemorrhagic disease in domestic pigs and wild boars. Since 2007, a highly virulent genotype II strain has emerged and spread in Europe and South-East Asia, where millions of animals succumbed to the disease. Field- and laboratory-attenuated strains of ASFV cause highly variable clinical disease severity and survival, and mechanisms remain unclear. We hypothesized that the immunological and hygienic status of pigs is a determinant of ASF disease course. Here we compared the immunological profile at baseline and in response to ASFV infection in specific pathogen-free (SPF) and farm-raised Large White domestic pigs. At steady state, SPF pigs showed lower white blood cell counts and a lower basal inflammatory and antiviral transcriptomic profile compared to farm pigs, associated with profound differences in gut microbiome composition. After inoculation with a highly virulent ASFV genotype II strain (Armenia 2008), severe clinical signs, viremia and pro-inflammatory cytokines appeared sooner in SPF pigs, indicating a reduced capacity to control early virus replication. In contrast, during infection with an attenuated field isolate (Estonia 2014), SPF pigs presented a milder and shorter clinical disease with full recovery, whereas farm pigs presented severe protracted disease with 50% lethality. Interestingly, farm pigs showed higher production of inflammatory cytokines, whereas SPF pigs produced more anti-inflammatory IL-1ra early after infection and presented a stronger expansion of leukocytes in the recovery phase. Altogether, our data indicate that the hygiene-dependent innate immune status has a double-edge sword impact on immune responses in ASF pathogenesis. While the higher baseline innate immune activity helps the host in reducing initial virus replication, it promotes immunopathological cytokine responses, and delays lymphocyte proliferation after infection with an attenuated strain. Such effects should be considered for live vaccine development and vigilance.

files are available from the EMBL-EBI European Nucleotide Archive with accession numbers E-MTAB-11634 and PRJEB52048.

**Funding:** This work was funded by grants of the Swiss Federal Food Safety and Veterinary Office (grant numbers 1.19.02 and 1.21.12) to NR, CB, and AS; and by internal funds of the Institute of Virology and Immunology (IVI). This research was made possible by funding from ICRAD, an ERA-NET co-funded under European Union s Horizon 2020 research and innovation programme (https://ec.europa.eu/programmes/horizon2020/en), under Grant Agreement n°862605. The funders had no role in study design, data collection and analysis, decision to publish, or preparation of the manuscript.

**Competing interests:** The authors have declared that no competing interests exist.

## Author summary

African swine fever virus (ASFV) causes a lethal hemorrhagic fever in domestic pigs and wild boars. The current epizootic of genotype II ASFV has recently affected hundreds of millions of animals on the Eurasian continent with major socio-economic impacts. Here, we found that differences in the baseline activation status of the immune system in specific pathogen-free (SPF) and conventional farm pigs are important determinants of ASF disease course. After inoculation with the highly virulent genotype II ASFV strain Armenia 2008, severe clinical signs, viremia and pro-inflammatory cytokines appeared sooner in SPF pigs than in farm pigs, indicating a reduced capacity to control early virus replication. In contrast, during infection with the attenuated strain Estonia 2014, SPF pigs presented a milder and shorter clinical disease with full recovery, whereas farm pigs presented a protracted disease with 50% lethality. Immunophenotyping, cytokines, blood transcriptome, and microbiota analyses before and during infection suggest that a higher baseline immune activation shown in farm pigs acts as a double-edge sword with consequences on disease severity and immune responses. These findings are of high importance in the context of deployment of live attenuated vaccines against ASF and more generally in understanding host-pathogen interactions in hemorrhagic fevers.

## Introduction

African swine fever virus (ASFV) causes a lethal hemorrhagic disease in domestic pigs and wild boars, and was first described in Kenya in 1921 [1]. ASFV is endemic in sub-Saharan Africa circulating quasi-asymptomatically in other members of the Suidae family such as warthogs and bushpigs, causing sporadic outbreaks in domestic pig farms. ASFV also replicates in soft ticks of the *Ornithodoros* genus found in Africa contributing to the sylvatic cycle of infection [2]. The first major epidemic of ASFV in pigs and wild boars was described in Europe, Russia and South America in the 1950s. This epidemic was caused by a genotype I strain that was eventually eradicated in the mid-1990s except in Sardinia, where an attenuated form of the disease remains endemic [3]. In 2007, a global genotype II ASFV epidemic started in the Caucasus region and spread from Georgia to neighboring countries by transmission in wild boars and further north by human activities up to Russia and Eastern Europe, where it continues to expand regionally in wild boar populations with occasional outbreaks in pig farms [4]. The same ASFV strain reached South East Asia in 2018 leading to the loss of millions of domestic pigs [5]. The most recent outbreaks in the Caribbean [6] and Italy, which are not directly adjacent to known foci of infection, highlight the risk of a worldwide spread of ASFV aided by long distance transport and release of contaminated animals or pork products. The lack of prophylactic and therapeutic solutions and the extreme stability of the virus in contaminated meat products and fomites are major challenges for animal welfare, economic hardship, food supply, and veterinary science.

ASFV is a large double-stranded DNA virus classified as the single representative of the *Asfarviridae* family [7]. Following nasal, oral, or parenteral infection through direct contact with infected animals, food, or ticks, ASFV rapidly spreads systemically via the blood circulation. ASFV has a narrow tropism for macrophages and high viral loads are found in blood and most organs, particularly in lymphoid organs [8, 9]. Currently circulating genotype II ASFV strains in Europe and Asia are virtually identical to the 2007 strain identified initially in Georgia. Upon infection, these strains cause sudden death or an acute disease with high fever and

apathy followed by skin hyperemia with petechial hemorrhages, hemorrhagic diarrhea, posterior paresis, cough, with death within 7–10 days [10]. Attenuated genotype II strains have more recently been isolated in wild boar in Eastern Europe causing a wider range of disease course, including mild to moderate disease [11–13]. Recent research efforts have focused on understanding the function of various ASFV genes implicated in virulence and control of innate immune responses [14–19]. However, the importance of the basal immune and hygienic status of the host for the antiviral response and the severity of the disease remain poorly understood. Here we characterized the immune and microbiota profiles of pigs raised in a conventional farm and in a specific pathogen-free (SPF) facility and compared their response to infection with two ASFV field strains of different virulence. We demonstrate that the more naïve immune system of SPF pigs results in a reduced control of virus replication at early stages of infection with both strains. However, both types of pigs succumbed similarly and rapidly following infection with the virulent Armenia 2008 strain. In contrast, after inoculation with the attenuated Estonia 2014 strain, SPF pigs showed a clear reduction of the severity and duration of clinical symptoms, whereas farm pigs developed a severe disease with 50% lethality associated with a severe cytokine storm.

## Results

### Immunological status of SPF pigs at steady state

To evaluate the effects of environmental exposures on the basal immune status of pigs, we compared hematological parameters of groups of Large White pigs raised in a conventional farm or in an SPF facility. At steady state, SPF pigs showed significantly higher platelet (PLT) and red blood cell (RBC) counts, and significantly lower white blood cell (WBC) counts, which were on average 50% lower than those of farm pigs (Figs 1A and S1A). Flow cytometry analysis of leukocyte subsets revealed that reduced numbers of polymorphonuclear neutrophils (PMNs) and monocytes accounted for the bulk of the leukocyte defect in the blood of SPF pigs (Figs 1B, S1B and S2). In addition, total $CD3^+$ T cells were also significantly reduced in SPF pigs, principally due to significantly reduced cytotoxic ($CD4^{neg}CD8^{hi}$) T lymphocytes and, for some animals, lower memory ($CD4^+CD8^+$) T helper (Th) cells. In contrast, SPF pigs had higher numbers of $CD4^{neg}CD8^{neg}$ T cells, representing a subset of TCR-γδ T cells, compared to farm pigs. To analyze further the differences between the two groups, we performed bulk RNA sequencing of whole blood and compared gene expression using blood transcriptional module (BTM) analysis [20]. The innate immune system of farm pigs had a higher level of activation as demonstrated by higher expression of BTM related to DC activation, IFN type I responses, inflammation, as well as BTM reflecting increased levels of circulating myeloid cells (Fig 2A). Also many BTM relating to cell cycle were overexpressed in farm pigs relative to SPF pigs pointing to ongoing adaptive immune cell activation (Fig 2B). Together, the flow cytometry and transcriptomic immunophenotyping establish that the farm-raised animals present a higher level of basal immune activation than the SPF pigs with increased numbers of leukocytes and significantly higher proliferative and inflammatory/antiviral transcriptional profile.

The intestinal microbiota and host immune system interact via multiple regulatory loops. We performed 16S rRNA gene sequencing of stool samples of SPF and farm pigs at steady state. While the Shannon (alpha) diversity was comparable between SPF and farm pigs (Fig 3A), we observed profound differences in microbiota composition between the two groups (Fig 3B). At the bacterial family level, we found a high prevalence of Prevotellaceae (Bacteroidetes phylum) in the farm group and a more balanced ratio of Bacteroidetes (Muribaculaceae, Bacteroidaceae) and Firmicutes (Lachnospiraceae, Oscillospiraceae) in SPF pigs (Figs 3C and

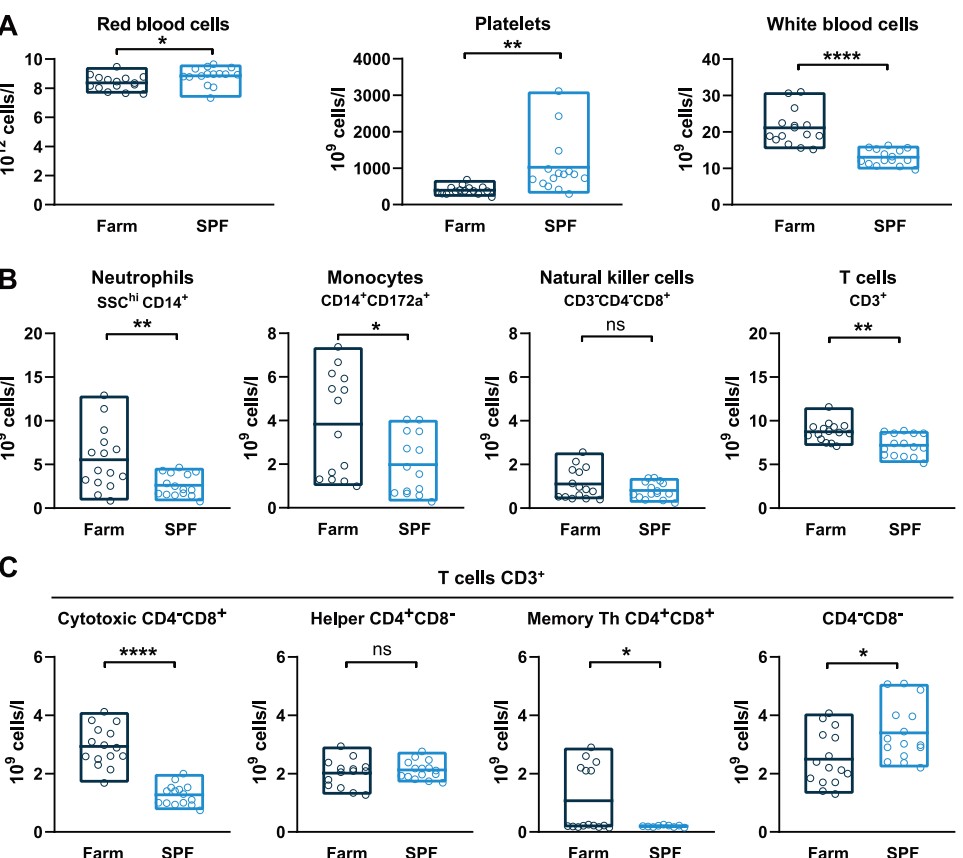

**Fig 1. Basal hematologic and immune profiles of SPF and farm pigs.** (A) RBC, PLT, and WBC counts in uninfected pigs. (B) Immunophenotyping of blood leukocytes (CD45+). The percentage of each subset was determined by flow cytometry gating (S2 Fig) and the absolute numbers were calculated using WBC counts. (C) T cell subsets gated from CD3+ T cells. (A, B, C) Each point represents the value for a single pig, horizontal lines and boxes represent the mean and range. Data are from 2 independent experiments (n = 15 per SPF or farm groups) and were analyzed using unpaired *t* test; ns, not significant; * $p < 0.05$; ** $p < 0.01$; **** $p < 0.0001$.

S3). Overall, housing and growth in a conventional farm enhances the basal activation status of the immune system and promotes the maintenance and growth of similarly diverse but distinct gut microbiome than pigs raised in SPF conditions.

## ASFV replication and disease after infection with the highly pathogenic Armenia 2008 strain

We next investigated whether the observed hematological and immune changes associated with farm or SPF housing altered the replication kinetics and the pathogenicity of a highly pathogenic ASFV strain (Armenia 2008). Six SPF and six farm pigs were inoculated intramuscularly with $6 \times 10^2$ TCID$_{50}$ of Armenia 2008. Viremia was detectable in 4 out of 6 SPF pigs and 2 out of 6 farm pigs at 1 dpi and in all animals from 2 dpi. Viremia was significantly higher at 4 dpi in SPF compared to farm pigs (Fig 4A). Accordingly, clinical signs such as fever, apathy, and loss of appetite appeared sooner in the SPF group (Fig 4B). One SPF pig was found dead on 6 dpi, two other SPF pigs were euthanized on this same day, and the 3 remaining SPF pigs were euthanized on 7 dpi, whereas all farm pigs were euthanized on 7 dpi because they were either close to or had reached the discontinuation criteria (Fig 4B). Macroscopic pathology at necropsy showed typical signs of acute ASFV infection as described previously [21], including

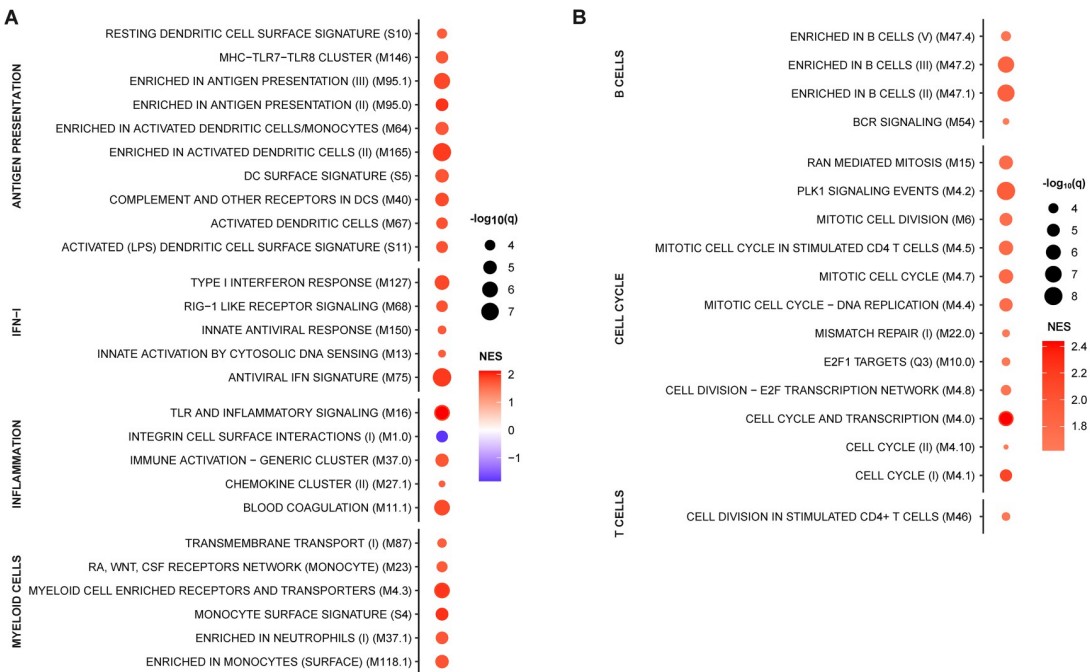

**Fig 2. Whole blood transcriptome module (BTM) analysis of farm pigs compared to SPF pigs.** Data points show significantly different (A) innate and (B) adaptive immune cell BTMs between farm and SPF pigs using SPF leukocytes as reference in steady state. BTM modulations were calculated as normalized enrichment scores (NES) using GSEA. Increased (red) or decreased (blue) BTMs in farm compared to SPF pigs are shown with color intensity proportional to the NES and the size of the data points proportional to the statistical q value. Data is from n = 12/group from 2 independent experiments.

splenomegaly, hemorrhagic lymph nodes, petechiae on kidneys, but no quantifiable differences were observed between the two groups. Virus loads were high (up to $10^8$–$10^9$ gEq/mg) in all organs of animals of both groups and especially in primary and secondary lymphoid organs such as tonsils, lymph nodes, spleen, and bone marrow. SPF pigs had significantly higher loads in all organs except bone marrow and submandibular lymph nodes (Fig 4C).

### The early innate immune response is increased in SPF versus farm pigs after Armenia 2008 infection

Consistent with the hemorrhagic nature of ASF, infection with Armenia 2008 led to a reduction in RBC, hematocrit and hemoglobin in both groups at 7 dpi (Figs 4D and S4A). PLT counts in SPF pigs were higher than the reference range at steady state and showed a slow decrease over time without developing an overt thrombocytopenia after infection. PLT counts remained within normal range in farm pigs throughout the infection (Fig 4D). A significant reduction in leukocyte numbers was observed at 2 dpi in SPF and at 4–5 dpi in farm pigs compared to baseline followed by a rebound (Fig 4D). Flow cytometry analysis revealed a general decrease in all blood leukocytes subsets with similar trends in farm and SPF pigs, except for monocytes, which were significantly increased at 5–7 dpi in SPF pigs (Figs 4E, 4F and S4B).

The cytokine response in serum was evaluated 1, 2, 4, 5 and 6 dpi. IFN type I bioactivity in serum was not statistically different between SPF and farm pigs. We observed a significantly

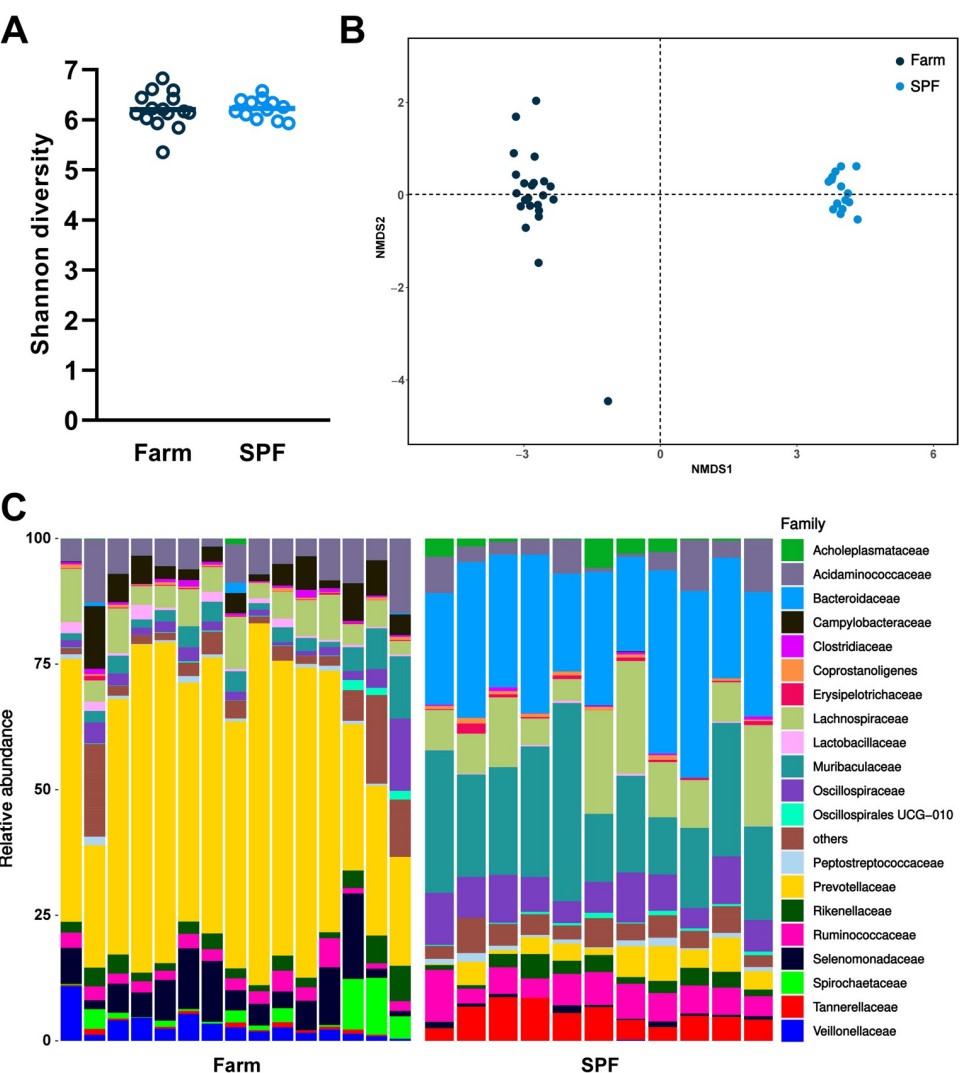

**Fig 3. Fecal microbiota composition at steady state in farm and SPF pigs.** (A,B) Shannon diversity and non-metric multidimensional scanning (NMDS) ordination of gut microbiome of pigs. Each data point represents the value for one animal. (C) Relative abundance of the most frequent bacterial families. Data are from 2 independent experiments (farm group n = 15; SPF group n = 11).

higher and earlier surge in pro-inflammatory IL1α, IL-1β, and IL-6 in serum of SPF pigs. Interestingly, anti-inflammatory IL-1ra and IL-10 were also significantly higher in SPF pigs (Fig 5B). In contrast, neutrophil chemoattractant chemokine IL-8 was expressed at higher levels in farm pigs (Fig 5B).

Whole blood transcriptomic module analysis revealed in both groups a highly significant induction of genes associated with IFN type I responses, antigen presentation, and cell cycle at 4 and 7 dpi compared to their respective transcriptomic profile prior to infection (Fig 6A). Only for some of these innate BTM, such as M165 (enriched in activated dendritic cells), M16 (TLR and inflammatory signaling) and some myeloid cell BTM, a higher induction was observed in the SPF group. Nevertheless, for the BTM belonging to the adaptive immune system, a more prominent downregulation of B cell BTM was observed clearly in the SPF group at early time points (Fig 6B). Furthermore, only in the SPF animals, T cell BTM were

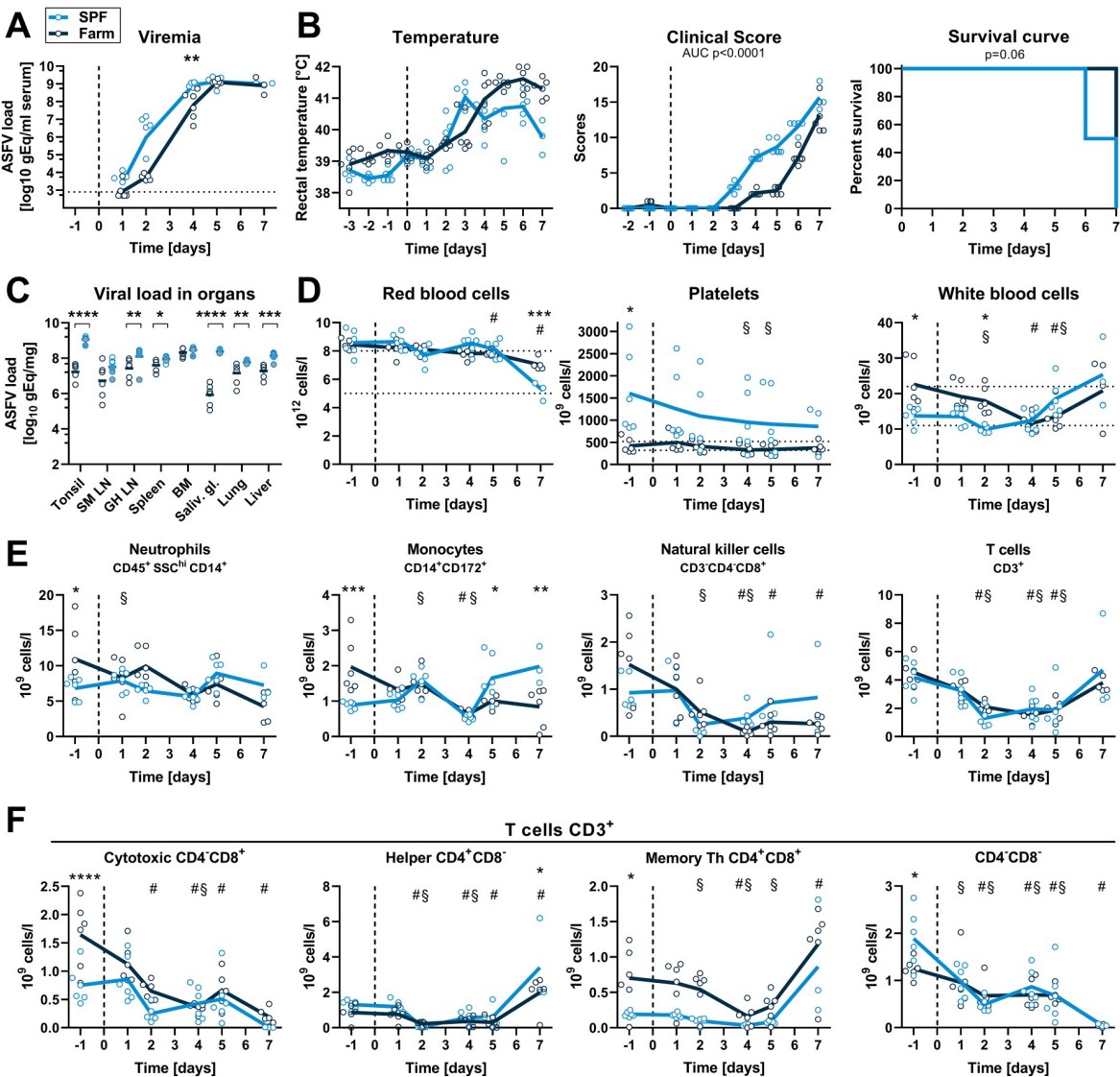

**Fig 4. Virus load, clinical signs and blood cell profiles after infection with virulent ASFV strain Armenia 2008.** SPF and farm pigs were inoculated intramuscularly with ASFV Armenia 2008. Blood samples were taken 1 day before infection and 1, 2, 4, 5, and 7 dpi. (A) Viremia was determined in serum by qPCR at indicated time points. (B) Rectal temperature, clinical score, and survival were reported daily. (C) Virus load in organs was measured by qPCR on the day of euthanasia (6 dpi: animal found dead, grey circles crossed (SPF n = 1), 6 dpi: grey circles (SPF n = 2), 7 dpi: empty circles (SPF n = 3, farm n = 6)). SM LN, submandibulary lymph node; GH LN, gastrohepatic LN; BM, bone marrow; Saliv. gl., salivary gland. (D) RBC, PLT, and WBC counts in blood at indicated dpi. (E) Immunophenotyping of blood leukocytes (CD45$^+$). Percentage of each subset was determined by flow cytometry gating (S2 Fig) and absolute numbers were calculated using WBC counts. (F) T cell subsets were gated from CD3$^+$ T cells. (A-F) Each data point represents the value for one pig, lines indicate the mean of each group. Data are from a single experiment (n = 6 pigs/group, except 7 dpi, where n = 3 for SPF group). (A, C-F) Differences between SPF and farm groups were analyzed by unpaired t test at each dpi with Holm-Sidak's correction for multiple comparisons. * p<0.05; ** p<0.01; *** p<0.001; **** p<0.0001. (B) Differences between groups for body temperature and clinical scores were analyzed by comparing the area under the curve (AUC). Differences in survival were analyzed by Log-rank (Mantel-Cox) analysis. (D-F) Significant differences at different dpi compared to the respective baseline (-1 dpi) are indicated for SPF (§, p<0.05) and farm (#, p<0.05); data were analyzed using mixed model analysis followed by Dunnett's multiple comparison.

downregulated at 4 dpi and some cell cycle BTM upregulated at 4 and 7 dpi. Taken together, this confirms the more prominent early perturbation of the immune system in the SPF animals, affecting particularly the peripheral lymphocyte compartment.

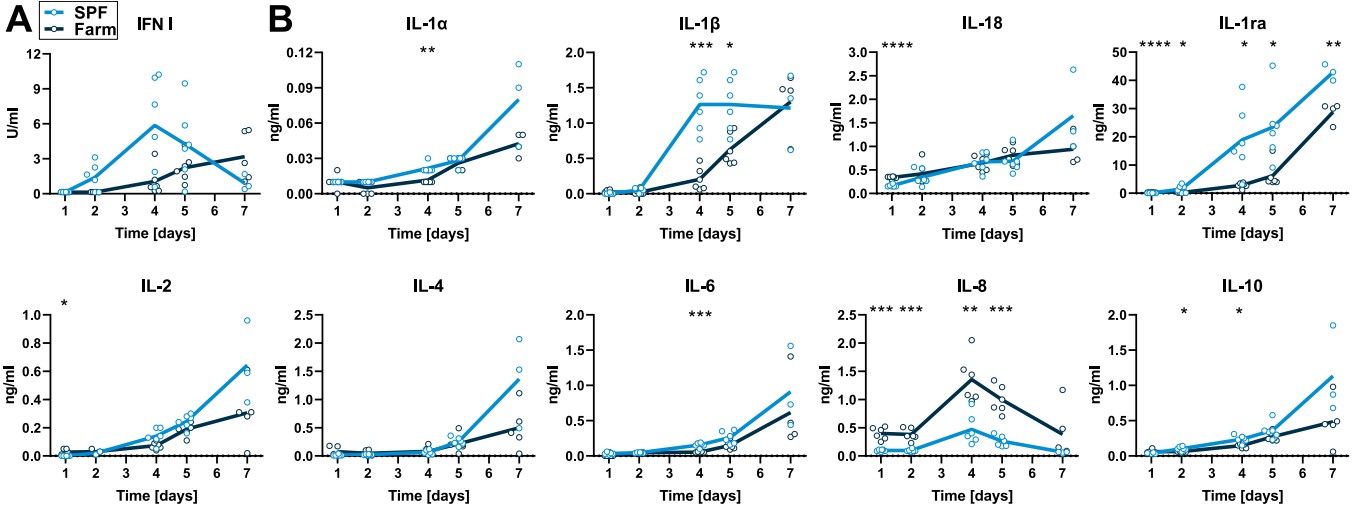

**Fig 5. Serum cytokines after infection with virulent ASFV Armenia 2008.** Blood samples were taken on 1, 2, 4, 5, and 7 dpi. (A) IFN type I activity in serum was measured by bioassay. (B) Cytokine levels were determined by multiplex ELISA. Data points show values for individual animals and lines indicate mean of each group at indicated dpi. Differences between SPF and farm groups were analyzed by unpaired t test at each dpi with Holm-Sidak's correction for multiple comparisons (n = 6 pigs/group, except 7 dpi, where n = 3 for SPF group). * p<0.05, ** p<0.01, *** p<0.001, **** p<0.0001.

## The clinical disease is more severe in farm than in SPF pigs infected with the attenuated ASFV Estonia 2014 strain

We then studied the clinical and immune response of groups of 6 farm and SPF pigs following infection with the field-attenuated ASFV strain Estonia 2014. As with the Armenia 2008 infection, the viremia reached maximal levels ($\sim 10^8$ gEq/ml serum) earlier in SPF pigs (peak 4 dpi) compared to farm pigs (peak 5 dpi), but viremia remained similarly high in both groups until the end of the study on 27 dpi (Fig 7A). The onset of clinical symptoms and the rise in body temperature was also detected one day earlier in SPF compared to farm pigs, correlating with the level of viremia. However, SPF pigs presented a significantly milder clinical disease in intensity and duration, and all SPF pigs recovered from infection. In contrast, farm pigs showed higher and prolonged fever and clinical scores from 5–18 dpi (Fig 7B). Notably, 3 of the 6 farm pigs, but none of the SPF pigs, were euthanized in the second week post infection– one pig on day 10 and two pigs on day 16 pi–because they reached the predetermined discontinuation criteria (Fig 7B). At necropsy, these animals had enlarged, hemorrhagic lymph nodes and petechial hemorrhages in the kidney, compared to the rest of the animals that had no gross lesions at the end of the study (28 dpi). Seroconversion was detectable in all farm pigs on 9, 11 and 14 dpi. In SPF pigs, seroconversion was significantly delayed and only detectable on 21 and 28 dpi, with values comparable to those of the farm pigs on these late time points (Fig 7C). Virus load in organs was similar in farm and SPF pigs that survived up to 27 dpi. In the farm pigs euthanized prematurely (10 and 16 dpi), the virus load was higher in spleen, lung, liver, and lymph nodes (S5A Fig).

Red blood cell counts, hematocrit and hemoglobin levels followed a downward trend over time after infection compared to baseline and counts were significantly lower in farm pigs from 1–21 dpi (Figs 7D and S5B). Platelet counts also decreased following infection in all animals, but severe clinical thrombocytopenia was principally observed in farm pigs (Fig 7D). ASFV induced a sharp drop in white blood cell counts by 4 dpi in both groups followed by a recovery (Fig 7D). Flow cytometry analysis of leukocyte subsets showed that ASFV induced a rapid depletion of all cell types (Figs 7E and S5C). Interestingly, neutrophil, monocyte and

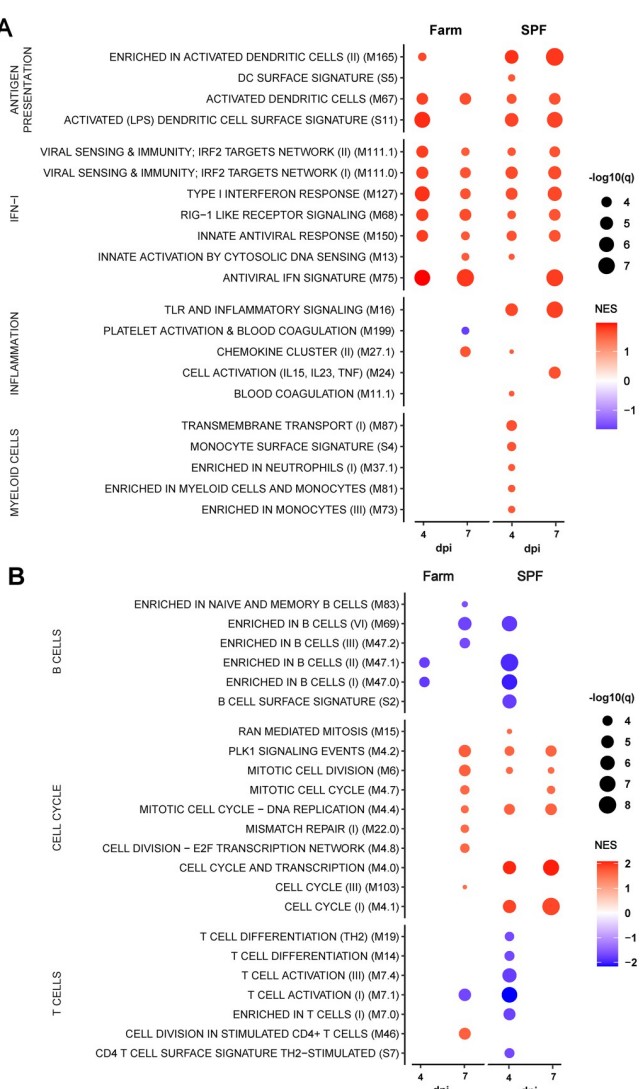

**Fig 6. BTM analysis after infection with virulent ASFV Armenia 2008.** Data shows only the significantly different (A) innate and (B) adaptive immune cell BTMs in farm and SPF pigs at 4 and 7 dpi relative to their respective BTM at baseline (day 0). BTM modulations were calculated as normalized enrichment scores (NES) using GSEA. Increased (red) or decreased (blue) BTMs are shown relative to each group's respective baseline with color intensity proportional to the NES and the size of the data points proportional to the q value. Data is from n = 6/group, except for 7dpi where n = 3 for SPF group.

CD8 T cell counts, which were all significantly lower in SPF pigs prior to infection, were comparable to the farm pigs 28 dpi. Similarly, CD4$^+$CD8$^+$ memory Th cell counts, while always significantly lower in SPF compared to farm pigs, doubled in SPF pigs at the end of the experiment compared to levels prior to infection (Figs 7F and S5C).

IFN type I activity in serum peaked on 4 dpi in both groups and no significant differences were found (Fig 8A). Pro-inflammatory cytokines IL-1α, IL-18, IL-2, IL-4 and IL-6 and anti-inflammatory IL-10 peaked at 7 dpi and levels were all significantly higher in farm compared to SPF pigs. The chemokine IL-8 peaked at 4–5 dpi and was also significantly higher in farm

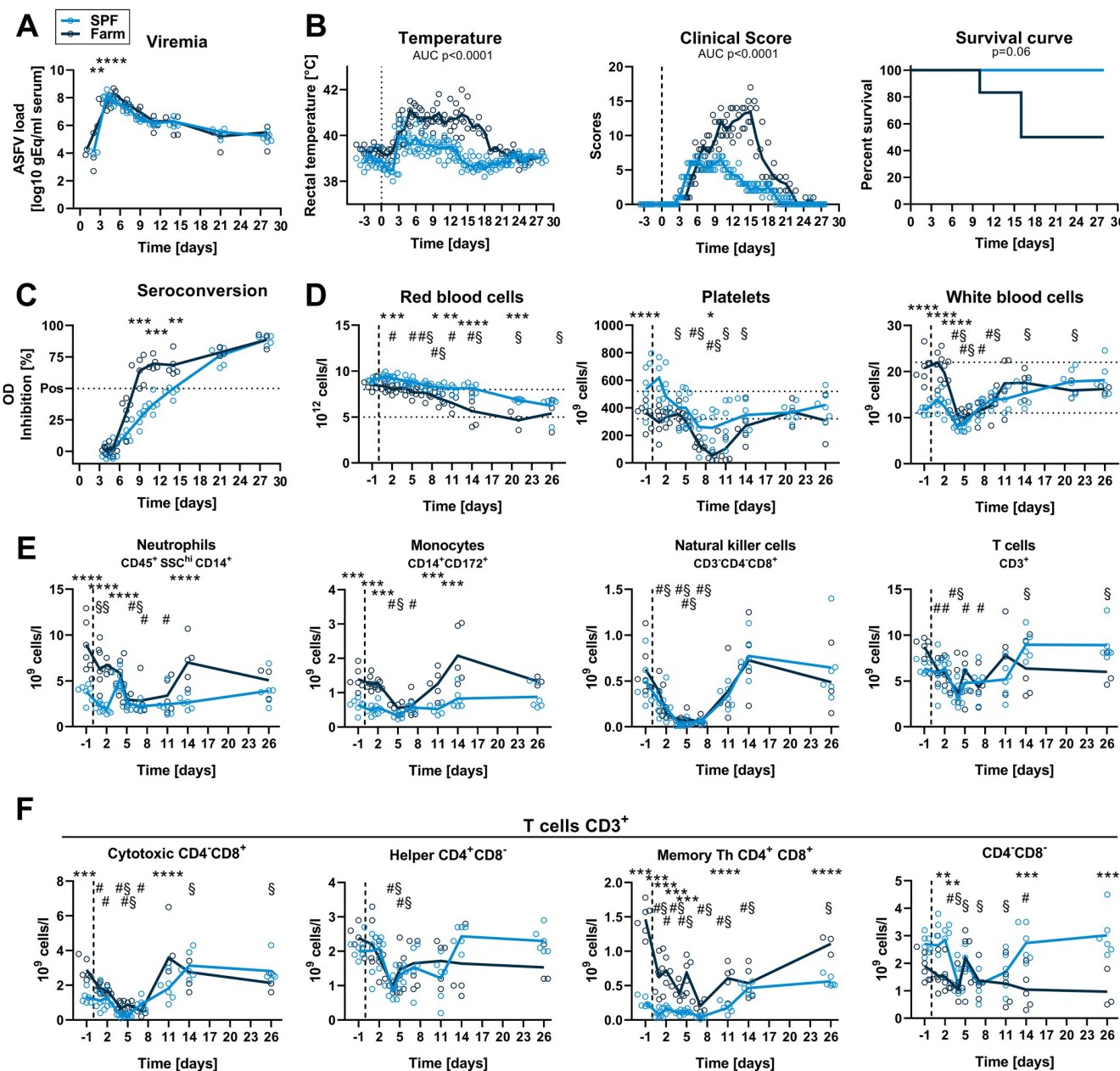

**Fig 7. Virus load, clinical signs, and blood cell profiles after infection with attenuated ASFV strain Estonia 2014.** SPF and farm pigs were inoculated intramuscularly with ASFV Estonia 2014 and blood samples were taken 1 day before infection and 1, 2, 4, 5, 7, 11, 14 and 26 dpi. All surviving animals were euthanized on 28 dpi. (A) Viremia was determined in serum by qPCR. (B) Rectal temperature, clinical score, and survival were reported daily. (C) Seroconversion was tested by ELISA. (D) RBC, PLT, and WBC counts in blood at indicated dpi. (E) Immunophenotyping of blood leukocytes (CD45$^+$). Percentage of each subset was determined by flow cytometry gating (S2 Fig) and absolute numbers were calculated using WBC counts. (F) T cell subsets were gated from CD3$^+$ T cells. (A-F) Data points represent values for individual pigs, lines indicate the mean of each group. Data are from a single experiment (n = 6 pigs/group, except from 11 dpi (n = 5) and from 15 dpi (n = 3) for farm group). (A, C-F) Differences between SPF and farm groups were analyzed by unpaired t test at each dpi with Holm-Sidak's correction for multiple comparisons. * p<0.05; ** p<0.01; *** p<0.001; **** p<0.0001. (B) Differences between groups for body temperature and clinical scores were analyzed by comparing the area under the curve (AUC). Differences in survival were analyzed by Log-rank (Mantel-Cox) analysis. (D, F) Significant differences in blood cell subsets at different dpi compared to the respective baseline (-1 dpi) are indicated for SPF (§, p<0.05) and farm (#, p<0.05); data were analyzed using mixed model analysis followed by Dunnett's multiple comparison.

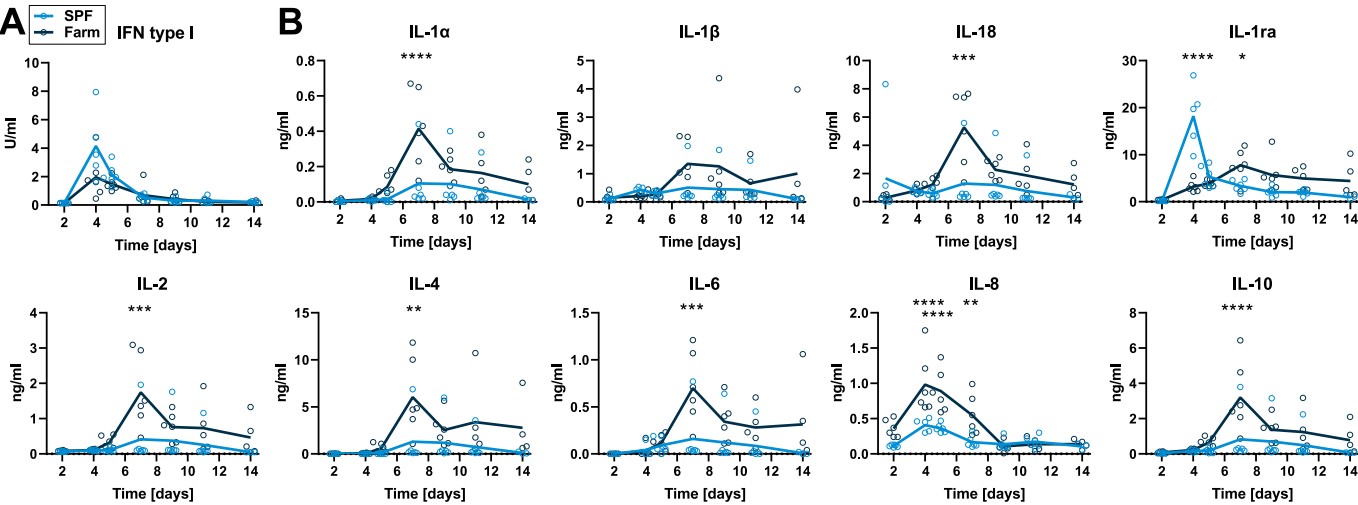

**Fig 8. Serum cytokines after infection with attenuated ASFV Estonia 2014.** Blood samples were taken on 2, 4, 5, 7, 9, 11, and 14 dpi. (A) IFN type I activity in serum was measured by bioassay. (B) Cytokine levels were determined by multiplex ELISA. Data points show values for individual animals and lines indicate mean of each group at indicated dpi. Differences between SPF and farm groups were analyzed by unpaired t test at each dpi with Holm-Sidak's correction for multiple comparisons (n = 6 pigs/group, except on 14 dpi, where n = 5 for farm group). * p<0.05, ** p<0.01, *** p<0.001, **** p<0.0001.

pigs (Fig 8B). Most interestingly, the natural IL-1 family antagonist IL-1ra peaked early at 4 dpi in SPF pigs (Fig 8B). These changes suggest that the more severe disease course in farm pigs may be driven by an overwhelming host inflammatory response.

Transcriptome analysis of whole blood on 4, 7 and 11 dpi show strongly induced innate immune BTM indicative of early DC activation and a sustained IFN type I signature (Fig 9A). Again, these responses did not appear to differ between the two groups. Also, platelet and blood coagulation BTM were downregulated at 7 dpi in both groups. For the adaptive BTM, we observed an early downregulation (4 dpi) of cell cycle, B and T/NK cell BTM, which affected more B cells and NK cell BTM in the farm pigs. The most prominent difference between farm and SPF pigs was however at 7 and 11 dpi for the cell cycle modules that were clearly more numerous and strongly induced in the SPF pigs (Fig 9B).

Taken together, the results with the attenuated ASFV Estonia 2014 strain suggest that the lower baseline immune activation of SPF pigs promotes a more resilient phenotype upon infection characterized by reduced inflammatory cytokine storm and early induction of anti-inflammatory IL-1ra, as well as a more prominent later induction of proliferative response during the adaptive phase of the antiviral response.

## Discussion

Our study shows profound differences in the baseline immune status of Large White pigs from the IVI's SPF facility compared to pigs from a conventional Swiss farm. The genetic background of the SPF colony is comparable to farm pigs in Switzerland, as semen is obtained from a unique source (SUISAG, Sempach, Switzerland). Furthermore, the MHC class I haplotype sequencing indicates that the SPF pigs from the IVI have swine leukocyte antigen (SLA) haplotype profiles with alleles commonly found in European Large White pig farms recently described [22]. The baseline immune and microbiota data was from two independent experiments at several months interval and was consistent for each facility prior to the start of each infection experiment. The white blood cell counts of the SPF pigs were in the lower range of normal values and were significantly lower than in the farm pigs. This difference was due to

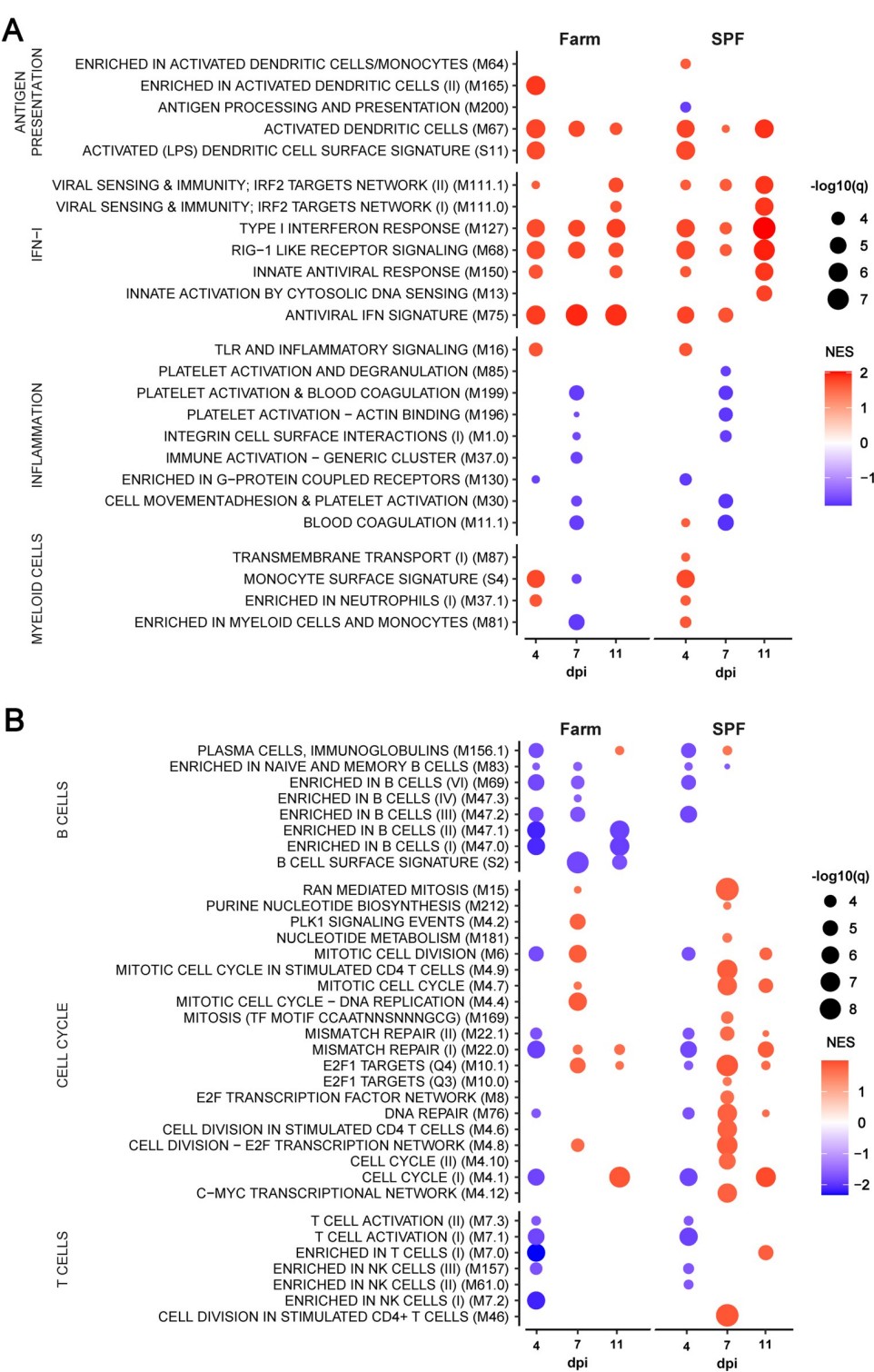

**Fig 9. BTM analysis after infection with attenuated ASFV Estonia 2014.** Data shows only the significantly different (A) innate and (B) adaptive immune cell BTMs in farm and SPF pigs at 4, 7 and 11 dpi relative to their respective BTM at baseline (day 0). BTM modulations were calculated as normalized enrichment scores (NES) using GSEA. Increased (red) or decreased (blue) BTMs are shown relative to each group's respective baseline with color intensity proportional to the NES and the size of the data points proportional to the q value. Data is from n = 6/group.

reduced numbers of neutrophils, monocytes, and most T cell subsets in SPF pigs, except CD4⁻CD8α⁻ T cells, which are principally composed of γδ T cells and were higher in SPF pigs. Of note, terminally differentiated effector γδ T cells are also present in the CD4⁻CD8α⁺ T cell gate and this specific subset could not be distinguished here from cytotoxic αβ T cells. We did not investigate if target cells of ASFV such as macrophages and dendritic cells, which are in part derived from circulating monocytes and in part derived from fetal precursors are also reduced in tissues of SPF pigs before and during infection. However, the faster appearance of viremia in SPF pigs compared to farm pigs inoculated with either virus suggests that a potential lack of target cell numbers in SPF pigs is not a likely factor influencing the initial viral replication kinetics of ASFV *in vivo*.

Importantly, whole blood transcriptomic profiling demonstrated a higher activation baseline for innate immune responses in farm pigs, notably the antiviral IFN signature, inflammatory signaling, antigen presentation activation and cell cycle. Thus, the unrestricted environmental exposure to various microbes in conventional farms appears to increase the alertness of the immune system compared to the more restricted exposure of SPF pigs. The analysis of the fecal microbiota also revealed profound differences in the composition of the microbiota between the two groups. As shown in wild, petshop, and SPF laboratory mice [23–26], exposure to pathogens and other microbes likely impacts the baseline activation of the pig immune system, which in turn may alter immune and inflammatory responses to infectious agents.

Experimental infection studies using virulent genotype II ASFV strains in domestic pigs have invariably shown severe disease with almost 100% lethality across various genetic backgrounds and hygiene conditions in domestic pigs and wild boars [27–30]. Accordingly, upon infection with the highly pathogenic Armenia 2008 strain, we found that both SPF and farm pigs develop a severe acute disease and 100% lethality. However, SPF pigs demonstrated an initial defect in controlling viremia compared to farm pigs. Furthermore, higher virus loads were found in most organs of SPF pigs at euthanasia 5–7 dpi. In the light of the baseline BTM profile of farm versus SPF pigs (Fig 2), these findings suggests that the immune activation baseline such as type I IFN and inflammation-related gene networks provides a protective barrier in the very early phase of infection.

Upon infection with the moderately virulent Estonia 2014 strain, a similar initial defect in containing the virus was observed in SPF pigs with a 24 h shift in the appearance of clinical symptoms and viremia. Surprisingly, the symptomatic disease course was now significantly shorter and less severe in SPF compared to farm pigs. Moreover, half of farm pigs were close to or reached the ethical criteria for termination of the experiment in the second week after infection, whereas all SPF pigs survived and did not show a typical bi-phasic disease. In line with our results, initial characterization studies showed that Estonia 2014 was highly pathogenic in wild boars, and less pathogenic in minipigs and domestic pigs [12, 14], indicating that genetics, housing and microbial exposure have an impact on the course of moderately virulent ASF. Specific-pathogen free (SPF) status of domestic pigs was previously associated with an acute, lethal disease course upon infection with the cell-culture passaged E75CV1 genotype I strain of ASFV that was shown to be attenuated in commercial domestic pigs [31]. Similarly, successive inoculation with the OURT88/3 and OURT88/1 attenuated genotype I strains caused clinical symptoms and death more frequently in SPF pigs than in farm pigs [32]. The results of these two studies appear to be at odds with our findings that SPF pigs are less susceptible to severe disease caused by the genotype II Estonia 2014 ASFV strain. The different results may be due to intrinsic differences of genotype I and II strains, or more specifically, caused by different gene deletions and associated mechanisms of attenuation. Moreover, differences in clinical course and immune responses of SPF pigs after ASFV inoculation are likely caused by specific

host immune and environmental factors such as the microbiota. SPF status is principally the exclusion of known porcine pathogens (defined in S1 Table for our study) and is not uniform in all facilities, especially regarding baseline immune status and bacterial commensals. We found that the microbiota and baseline immune status are very stable over time in SPF pigs from the IVI facilities and in pigs obtained from the same commercial farm at several months interval. Specific immune cell phenotype, blood transcriptomic profile and microbiota identified here in each group may thus serve as a reference for future studies to better understand factors associated with disease severity and ultimately protective adaptive immune responses.

High clinical score and sustained fever in farm pigs was associated with higher levels of inflammatory cytokines at 7 dpi. In contrast, SPF pigs showed a high and early peak of the anti-inflammatory cytokine IL-1ra measured at 4 dpi. IL-1ra blocks IL-1R1 and prevents IL-1 signaling. High levels of IL-1ra early in the infection may contribute to down-regulate the induction of a cytokine storm in the later stage of infection that may be amplified by IL-1 family cytokines. A previous *in vitro* study of primary macrophages infected with the virulent Georgia 2007 ASFV strain has also shown that IL-1ra is one of the most transcriptionally upregulated cytokine genes from 6-18h post infection [33]. Further studies are needed to determine the impact of IL-1 family cytokines and particularly IL-1ra on ASFV immunopathology. SPF pigs also showed a more prominent induction of cell cycle in blood leukocytes at day 7 and 11 post infection. Whether this is an early sign of a more prominent T cell response will be of high importance for future studies. Furthermore, detailed flow cytometry studies of T cell responses to infection with ASFV genotype II strains Estonia 2014 and Armenia 2008 recently showed that severe disease is associated with impaired T cell responses in blood and tissues of domestic pigs and wild boars [10, 34]. These studies and previous seminal studies provide further impetus to investigate the functions of CD8$^+$ ($\alpha\beta$ and $\gamma\delta$) T cells and particularly perforin$^+$ cells in the pathogenesis and protective immune responses to ASFV [35–37].

Immune activation and cytokine responses in farm and SPF pigs may be determined by the different gut microbiota and their metabolites. Farm pigs in our study had very high levels of Prevotellaceae, which have been generally associated with better production and health in pigs [38]. Whether Prevotellaceae contribute to the higher basal immune status of farm pigs compared to SPF pigs with relatively lower Prevotellaceae remains to be established. Furthermore, different species of Prevotellaceae may have a different impact on the immune system and we have shown that a higher immune baseline may not necessarily be an advantage upon infection with an attenuated ASFV strain. Previous studies showed that fecal microbiota transplantation from resistant warthogs to domestic pigs may alter the susceptibility to an attenuated, but not to a virulent, genotype I ASFV strain [39]. While promising, caution should be taken in the interpretation as the microbiota composition following transplantation did not significantly differ between transplanted animals and control groups and the disease was generally mild in all groups [39, 40]. Further fecal transplantation studies are warranted to determine whether the gut microbiota composition has a significant role in ASFV pathogenesis and to identify key bacterial species and metabolites involved in modulating the pig immune system in a favorable manner.

Overall, our data demonstrate that the hygiene status-dependent baseline immune activity impacts viral pathogenesis and host antiviral responses in a virulence-dependent manner. On one side, a higher baseline innate immune activity helps the host in reducing initial replication of a highly virulent ASFV causing acute disease and early mortality. On the other side, a higher immune baseline has clear detrimental effects in terms of immunopathological inflammatory cytokine responses and delayed lymphocyte proliferation, when infection occurs with a less virulent virus causing acute to chronic ASF.

## Materials and methods

### Ethics statement

The study was performed in compliance with the Animal Welfare Act (TSchG SR 455), the Animal Welfare Ordinance (TSchV SR 455.1), and the Animal Experimentation Ordinance (TVV SR 455.163) of Switzerland. All experiments were reviewed by the committee on animal experiments of the canton of Bern and approved by the cantonal veterinary authority under the license BE18/2019.

### *In vivo* studies

A total of 30 male and female Large White domestic pigs, 10 to 11 week-old and with a body weight of approximately 20–25 kg were obtained from the IVI SPF breeding facility or a local farm. The pigs from the IVI SPF facility are not vaccinated and are negative for all porcine viral and bacterial pathogens listed in S1 Table. Access to the SPF facility is restricted to animal caretakers and includes showering airlock at entrance. The SPF facility is supplied with HEPA-filtered air at positive pressure and pigs are provided irradiated pig pellet diet and autoclaved hay and straw. No animal has been introduced in the facility since the initial colonization in 1993 and no disease clinical signs (diarrhea, abortion, respiratory symptoms) were observed during this period. The genetic pool is maintained at a similar status as current Swiss/European Large White pig farms via artificial insemination using semen from SUISAG (Sempach, Switzerland). The commercial farm pigs were from a high standard fattening pig farm and were vaccinated against porcine circovirus 2, *Escherichia coli*, and *Lawsonia intracellularis*. The SPF (n = 15) and farm pigs (n = 15) were moved to the BSL-3Ag containment facilities in two independent experiments and randomly assigned to separate groups of six or three animals, for virus- or mock-infection respectively. During the trials, the animals were fed a commercial pig pellet diet (Granovit AG, Kaiseraugst, Switzerland) with hay supplementation and water *ad libitum*. After 5 days of acclimatization, the pigs were infected by intramuscular injection of 2 ml DMEM containing $\sim$3 to $6\times10^2$ $TCID_{50}$/ml of either the highly virulent Armenia 2008, or the naturally attenuated Estonia 2014 strain. Mock-infected animals received 2 ml of DMEM. Body temperature and clinical parameters were assessed daily by a veterinarian (KM, NR, CB) based on an adapted clinical score checklist previously described for experimental classical swine fever virus infections [41]. All pigs were euthanized at the latest on day 28 post infection unless discontinuation criteria were reached. A cumulative score of 18 and/or a score of 3 in one of the following parameters; liveliness, body tension, breathing, walking or skin, were defined as the discontinuation criteria of the experiment. Blood samples were collected 1–3 days prior to infection and on 1, 2, 4, 5, 7, 9, 11, 14, 21, and 26 days post-infection, or until discontinuation criteria were reached. A full necropsy was performed on all animals; blood and organs were collected for virus quantification and titration.

### Virus stocks and quantification

The genotype II ASFV strains Armenia 2008 [42] and Estonia 2014 [12, 14] (Genbank accession number LS478113.1) were generously provided by Sandra Blome and Martin Beer, Friedrich-Loeffler-Institut, Greifswald–Insel Riems, Germany. Virus titration was determined by indirect immunofluorescence assay in WSL-R-HP cells [43], a kind gift from Matthias Lenk, Collection of Cell Lines in Veterinary Medicine, Friedrich-Loeffler-Institut, Greifswald–Insel Riems, Germany. Briefly, 96-well-plates were seeded with $1.5 \times 10^6$ cells in DMEM supplemented with 10% FBS, 50 U/ml penicillin-streptomycin and 2.5 µg/ml amphotericin B (ThermoFisher Scientific). Cells were infected 24 h later with ten-fold dilutions of organ

homogenates in five replicates. After 48 h, the cells were fixed with 4% neutral-buffered formalin for 10 min at room temperature, permeabilized and incubated with 1:40 dilution of convalescent serum (produced in-house) in 0.3% saponin, and a protein A–FITC conjugate (1:200 in 0.3% saponin) was used for visualization. Positive cells were counted and the titer was calculated using the Reed-Muench method. For qPCR, DNA was extracted using the NucleoMag VET kit (Macherey-Nagel) and the KingFisher extraction platform following manufacturers' instructions. All nucleic acid extractions were performed with 200 μl of either serum, whole blood, swab soaked in RA1 lysis buffer or organ homogenates in RA1 lysis buffer adjusted to contain 5 mg of tissue. Subsequently, qPCR was performed according to published protocol [44] with slight modifications. For quantification of genome equivalents (gEq), a 1946bp fragment of the B646L (p72) ORF containing the qPCR target sequence was cloned in the pCR4-TOPO Vector (ThermoFisher Scientific). Ten-fold dilutions of the linearized plasmid with known DNA-concentration were subsequently used to generate a standard curve from which the gEq of the samples were calculated. Samples and standards were run in triplicate.

## Hematology and flow cytometry

Differential blood cell counts were determined from EDTA blood samples using an automated hematology analyzer (VetScan HM5, Abaxis). The percentage of leukocyte subsets were determined by flow cytometry and absolute subset counts were calculated using WBC values from the hematology analyzer. Whole blood (100 μl) or single-cell suspensions ($1 \times 10^6$ cells) of spleen, tonsils, lymph nodes, and liver were incubated with antibody panels (Table 1). Data acquisition (100'000 single-cell events) was done on a BD FACS Canto II (BD Bioscience), and data analyzed with FlowJo v10.

## Cytokine and antibody measurements

Serum cytokines were determined using a custom premixed Milliplex Map porcine cytokine/chemokine magnetic beads kit (Millipore, USA) for 13 cytokines i.e. IFNγ, IL-1α, IL-1β, IL-1ra, IL-2, IL-4, IL-6, IL-8, IL-10, IL-12, IL-18, and TNF-α. The bioactivity of porcine IFN-α/β was determined using a bioassay as described previously [45]. Briefly, serum was heat inactivated and incubated with SK6-MxLuc cells. The luciferase activity was measured using a Luciferase reader (GloMax, Promega) and recombinant porcine IFN-β produced in HEK293T cells were used as standard. Antibodies against ASFV protein p72 were detected using INgezim

**Table 1. List of antibodies.**

| Fluorophore | Target | Species, Isotype | Source | Cat# | Clone | Final dilution |
|---|---|---|---|---|---|---|
| -- | CD3 | Mouse IgG1 | hybridomas | | PTT3 | 1/20 |
| -- | CD8 | Mouse IgG2a | hybridomas | | 76-2-11 | 1/4 |
| -- | CD4 | Mouse IgG2b | hybridomas | | 74-12-4 | 1/100 |
| -- | CD45 | Mouse IgM | hybridomas | | 74-9-3 | 1/2 |
| -- | CD172a | Mouse IgG2b | hybridomas | | 74-22-15A | 1/20 |
| PerCP-Cy5.5 | CD4 | Mouse IgG2b | BD | 561474 | 74-12-4 | 1/50 |
| FITC | CD14 | Mouse IgG2b | Coulter | 6603511 | 322A-1 My4 | 1/50 |
| BV421 | IgG2b | Rat IgG2b | BD | 743174 | II/41 | 1/200 |
| PE | IgG2b | Goat IgG2b | BioConcept | 0104–09 | A-1 | 1/100 |
| AF488 | IgG1 | Goat IgG | Thermo Fisher | A-21121 | | 1/1000 |
| AF647 | IgM | Goat IgG | Thermo Fisher | A-21238 | | 1/1000 |
| AF647 | IgG2a | Goat IgG | Thermo Fisher | A-21241 | | 1/1000 |

PPA COMPAC blocking ELISA Kit (R.11.PPA.K.3, Ingenasa, Madrid, Spain) according to manufacturer's instruction. The results are expressed in % of inhibition, with following cut-off values: <40%, negative; 40–50%, doubtful; ≥50%, positive. Serum samples were tested in duplicate.

## RNA-seq data analysis

Peripheral blood (2.5 ml) was collected into a PAXgene RNA tube (PreAnalytiX, Qiagen) from all pigs 2 days prior to infection and then on 4, 7 and 11 days post infection (dpi). Tubes were inverted 10 times after collection and incubated at room temperature for 4 h for RNA stabilization, transferred to -20˚C overnight, and stored at -80˚C. RNA extraction was performed using the Paxgene Blood RNA kit (Qiagen, Venlo, The Netherlands) and the RNA quality was controlled with a Fragment Analyzer (5200 Fragment Analyzer CE instrument, Agilent). All samples showed a high RNA quality (RNQ>8) and were sequenced using an Illumina HiSeq 3000 sequencer (Illumina, San Diego, CA, USA). Reads were mapped to the Sscrofa11.1 assembly of the Swine Genome Sequencing Consortium (SGSC) reference from the National Center for Biotechnology Information (NCBI) (https://www.ncbi.nlm.nih.gov/assembly/GCF_000003025.6/) using HISAT2v.2.1.0. Differential gene expression analyses between different time points, groups and viruses were performed on R using the Bioconductor package DESeq2 v. 1.18.1.

## Blood transcriptional modules (BTM) analyses

Differential gene expression was enriched and ranked using GSEA. For ranking, the negative natural logarithm of the P-values with genes upregulated (positive log2-fold change values) and the natural logarithm of the P-values with genes downregulated were calculated [46]. Significant gene perturbations (FDRqvalue<0.05) were presented using BTM [20] adapted to the pig genome by replacing human genes with their pig homologs [47]. Figures were created in R using ggplot2 Bioconductor package.

## Fecal microbiota analysis

Fecal samples were collected from SPF and farm pigs prior to infection, at least one week after acclimatization into the biocontainment stables of the IVI. Stool collection was performed using sterile swabs and containers and stored at -80˚C. DNA was extracted using the QIAamp Fast DNA Stool Mini Kit following the manufacturer's guidelines. For an optimal lysis of and separation of impurities from stool samples, the stools were first suspended and vortexed in 1 ml of InhibitEX Buffer. The V4 region of the 16S rRNA gene was amplified using forward (5'-GTGCCAGCMGCCGCGGTAA-3') and reverse (5'-GGACTACHVGGGTWTCTAAT-3′) primers modified with an Illumina adaptor sequence at the 5′ end. PCR products were purified using the QIAquick PCR Purification Kit (Qiagen, Hilden, Germany). Samples were passed through to a MiSeq Illumina sequencing platform for indexing and paired-end sequencing (2 × 250 bp; reagent kit, v2). Sequencing data were analyzed as previously described using the DADA2 package (version 1.16.0) in R software (version 4.0.2) for the identification of amplicon sequence variants (ASV). The taxonomy assignment of the ASVs was done using the SILVA (version 132) database. Contaminating sequences were identified using the decontam package (version 1.8.0) in R. Contaminants were identified by their frequency of occurrence and independently within each batch. Based on the ASVs identified using DADA2, we calculated the alpha-diversity values for Shannon diversity indices using the estimate_richness command in the phyloseq package (version 1.32.0) in R. The alpha-diversity values for the sample types (farm and SPF pigs) were calculated and analyzed statistically using Wilcoxon rank-sum

tests (using the wilcox.test function in R). Distance matrices were calculated for the beta-diversity analyses and used as input files for the non-metric multidimensional scaling (NMDS) plots as described [48]. Statistical analysis was performed by permutation test (PERMANOVA; Adonis function).

## Statistical analysis

Statistical analyses were performed using GraphPad Prism version 8.0.0 for Windows unless otherwise indicated. Number of samples and statistical tests used are indicated in figure legends.

## Supporting information

**S1 Fig. Basal hematologic profiles of SPF and farm pigs.** (A) Hematocrit, hemoglobin and mean corpuscular hemoglobin concentration (MCHC) counts in uninfected pigs. (B) Immunophenotyping of blood leukocytes (CD45$^+$). Percentage of each subset was determined by flow cytometry gating. (C) T cell subsets were gated from CD3$^+$ T cells. (A-C) Each point represents the value for a single pig, horizontal lines and boxes represent the mean and range. Data are from 2 independent experiments (n = 15 per SPF or farm groups) and were analyzed using unpaired t test; $^*$ p$<$0.05; $^{**}$ p$<$0.01; $^{***}$ p$<$0.001.
(PDF)

**S2 Fig. Flow cytometry gating strategies for blood leukocyte immunophenotyping.** (A, B) Representative flow cytometry dot plots of single cells with gates for total leukocytes (WBC, CD45$^+$), neutrophils (CD45$^+$SSC$^{hi}$CD14$^+$) and monocytes (CD45$^+$SSC$^{low}$CD14$^+$CD172a$^+$) of farm (A) and SPF (B) pigs measured at baseline. (C, D) Representative flow cytometry dot plots of leukocytes (WBC, CD45$^+$) with gates for T cell subsets (CD3$^+$) based on CD4 and CD8 markers and NK cells (CD3$^-$CD8$^+$) in farm (C) and SPF (D) pigs measured at baseline.
(PDF)

**S3 Fig. Abundance of the most frequent bacterial families present in fecal microbiota of farm and SPF pigs at steady state.** Animals from 2 independent experiments are included (farm group n = 15, SPF group n = 11).
(PDF)

**S4 Fig. Blood cell profiles after infection with virulent ASFV strain Armenia 2008.** SPF and farm pigs were inoculated intramuscularly with ASFV Armenia 2008. Blood samples were taken 1 day before infection and 1, 2, 4, 5, and 7 dpi. (A) Hematocrit, hemoglobin and mean corpuscular hemoglobin concentration (MCHC) counts in blood at indicated dpi. (B) Percentage of leukocyte (CD45$^+$). subsets in blood determined by flow cytometry (S2 Fig). Data points represent values for individual pigs, lines indicate mean of each group. Data are from a single experiment (n = 6 pigs/group, except 7 dpi, where n = 3 for SPF group). Differences between SPF and farm groups were analyzed by unpaired t test at each dpi with Holm-Sidak's correction for multiple comparisons. $^*$ p$<$0.05; $^{**}$ p$<$0.01; $^{***}$ p$<$0.001. (C) T cell subsets gated on CD45$^+$CD3$^+$.
(PDF)

**S5 Fig. Viral load and blood cell profiles after infection with attenuated ASFV strain Estonia 2014.** SPF and farm pigs were inoculated intramuscularly with ASFV Estonia 2014. Blood samples were taken 1 day before infection and 1, 2, 4, 5, 7, 11, 14 and 26 dpi. (A) Virus load in organs was measured by qPCR on the day of euthanasia (5 dpi grey circles crossed (SPF n = 1), 6 dpi, grey circles (SPF n = 2), 7 dpi, empty circles (SPF n = 3, farm n = 6)). SM LN,

submandibular lymph node; GH LN, gastrohepatic LN; BM, bone marrow; Saliv. gl., salivary gland. (B) Hematocrit, hemoglobin and mean corpuscular hemoglobin concentration (MCHC) values in blood. (C) Percentage of leukocyte (CD45$^+$). subsets in blood determined by flow cytometry (S2 Fig). (D) T cell subsets gated from CD3$^+$ T cells. Data are from a single experiment (n = 6 pigs/group, except from 11 dpi (n = 5) and from 15 dpi (n = 3) for farm group). (B-D) Differences between SPF and farm groups were analyzed by unpaired t test at each dpi with Holm-Sidak's correction for multiple comparisons. $^*$ p<0.05; $^{**}$ p<0.01; $^{***}$ p<0.001; $^{****}$ p<0.0001.
(PDF)

**S1 Table. List of pathogens excluded from SPF pigs.**
(DOCX)

## Acknowledgments

We thank Daniel Brechbühl, Katarzyna Sliz, Hans-Peter Lüthi, Jan Salchli, and Roman Troxler for excellent animal care. We thank Aurélie Godel, Sylvie Python, Markus Gerber, Obdulio Garcia-Nicolas, and Matthias Liniger for technical assistance and fruitful discussions. We thank Pamela Nicholson from the NGS platform, University of Bern for help with library preparation and RNA sequencing; Geert van Geest from the Interfaculty Bioinformatics Unit, University of Bern for help with sequencing data analysis; and Sabine Hammer from the Institute of Immunology, University of Veterinary Medicine Vienna for MHC haplotyping of our SPF herd. We are grateful to Imbi Nurmoja and Olev Kalda from the Estonian Veterinary and Food Laboratory, Tartu, Estonia for their agreement to transfer the Estonia 2014 isolate to the IVI, Switzerland. We are grateful to Sandra Blome and Martin Beer from Friedrich-Loeffler-Institute, Greifswald-Insel Riems, Germany, for sending the two ASFV strains Estonia 2014 and Armenia 2008 and for the precious collaboration.

## Author Contributions

**Conceptualization:** Artur Summerfield, Nicolas Ruggli, Charaf Benarafa.

**Data curation:** Emilia Radulovic, Kemal Mehinagic, Nicolas Ruggli, Charaf Benarafa.

**Formal analysis:** Emilia Radulovic, Kemal Mehinagic, Tsering Wüthrich, Markus Hilty, Artur Summerfield, Nicolas Ruggli, Charaf Benarafa.

**Funding acquisition:** Artur Summerfield, Nicolas Ruggli, Charaf Benarafa.

**Investigation:** Emilia Radulovic, Kemal Mehinagic, Horst Posthaus, Nicolas Ruggli, Charaf Benarafa.

**Methodology:** Emilia Radulovic, Kemal Mehinagic, Markus Hilty, Artur Summerfield, Nicolas Ruggli, Charaf Benarafa.

**Project administration:** Nicolas Ruggli, Charaf Benarafa.

**Resources:** Artur Summerfield, Nicolas Ruggli, Charaf Benarafa.

**Supervision:** Nicolas Ruggli, Charaf Benarafa.

**Validation:** Emilia Radulovic, Kemal Mehinagic, Markus Hilty, Artur Summerfield, Nicolas Ruggli, Charaf Benarafa.

**Visualization:** Emilia Radulovic, Tsering Wüthrich, Artur Summerfield, Charaf Benarafa.

**Writing – original draft:** Emilia Radulovic, Charaf Benarafa.

**Writing – review & editing:** Emilia Radulovic, Artur Summerfield, Nicolas Ruggli, Charaf Benarafa.

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
