## [Decision Letter · Decision Letter 0]

16 May 2022

Dear Prof. Dr. Benarafa,

Thank you very much for submitting your manuscript "The baseline immunological and hygienic status of pigs impact disease severity of African swine fever" for consideration at PLOS Pathogens. As with all papers reviewed by the journal, your manuscript was reviewed by members of the editorial board and by several independent reviewers. In light of the reviews (below this email), we would like to invite the resubmission of a significantly-revised version that takes into account the reviewers' comments.

The reviewers found the manuscript well written and that it addressed important issues directed at understanding susceptibility of pigs, of defined immune and health status, to virulent and an attenuated African swine fever virus (ASFV) isolate. Detailed results are presented which significantly extend previous studies regarding the immune status, microbiome and blood transcriptional responses in infected pigs. These results would be of benefit to the scientific community and more specifically to other researchers interested in pathogenesis of ASFV including evaluating of candidate vaccines. as well as more broadly.

Reviewers 1 and 2 felt that the data presented did not adequately support the conclusions drawn. Experiments are lacking to demonstrate that the measured immune responses in the SPF pigs and the farm pigs are the cause of the reported differences in response. One reviewer suggests additional experiments to support the conclusions. These reviewers also comment that previous publications relevant to your manuscript had not been cited and/or that differences between your results and previous results had not been adequately discussed.

The third reviewer has concerns about the gating strategy used in the FACS experiments. These concerns may be possible to address by further analysis of the data or considered in discussion. The reviewer also questioned the descriptions of use of the anti-CD45 antibody.

Any revision of your manuscript should address the points made by the reviewers.

We cannot make any decision about publication until we have seen the revised manuscript and your response to the reviewers' comments. Your revised manuscript is also likely to be sent to reviewers for further evaluation.

Sincerely,

Linda Kathleen Dixon

Associate Editor

PLOS Pathogens

Klaus Früh

Section Editor

PLOS Pathogens

Kasturi Haldar

Editor-in-Chief

PLOS Pathogens

orcid.org/0000-0001-5065-158X

Michael Malim

Editor-in-Chief

PLOS Pathogens

orcid.org/0000-0002-7699-2064

The reviewers found the manuscript well written and that it addressed important issues directed at understanding susceptibility of pigs, of defined immune and health status, to virulent and an attenuated African swine fever virus (ASFV) isolate. Detailed results are presented which significantly extend previous studies regarding the immune status, microbiome and blood transcriptional responses in infected pigs. These results would be of benefit to the scientific community and more specifically to other researchers interested in pathogenesis of ASFV including evaluating of candidate vaccines. as well as more broadly.

Reviewers 1 and 2 felt that the data presented did not adequately support the conclusions drawn. Experiments are lacking to demonstrate that the measured immune responses in the SPF pigs and the farm pigs are the cause of the reported differences in response. One reviewer suggests additional experiments to support the conclusions. These reviewers also comment that previous publications relevant to your manuscript had not been cited and/or that differences between your results and previous results had not been adequately discussed.

The third reviewer has concerns about the gating strategy used in the FACS experiments. These concerns may be possible to address by further analysis of the data or considered in discussion. The reviewer also questioned the descriptions of use of the anti-CD45 antibody.

Any revision of your manuscript should address the points made by the reviewers.

Reviewer's Responses to Questions

**Part I - Summary**

Reviewer #1: African swine fever (ASF) has become without any doubt, the number one threat for the swine industry worldwide.

ASF research has been traditionally neglected, considering it an “exotic” problem and today, being a global problem, there is not enough knowledge to rapidly counteract its progression.

In this aspect, the article here presented tackles a very important issue, such as it is, understanding the immunological mechanisms involved in susceptibility to ASFV.

On this regard, my first concern is that authors have ignored seminal findings describing that specific pathogen-free (SPF) pigs are much more susceptible to attenuated ASFV strains than conventional pigs (King et al., Vaccine. 2011 Jun 20; 29(28): 4593–4600; Lacasta et al. J Virol. 2014 Nov;88(22):13322-32.); exactly the opposite results here presented. Not mentioning these works in the introduction, from the very beginning, is somehow surprising but not citing them in the discussion is even more difficult to explain. Work performed by two research groups showed higher susceptibility of SPF pigs to two different attenuated strains of ASFV (OURT 88/3 and E75CV1), both of them form the Genotype I isolated in the Iberian peninsula during the endemic period (1957-1997).

I strongly recommend revising the entire manuscript taking into account this reality and the fact that SPF and conventional pigs used in these studies are most probably from different origins than the ones used in the present work.

I encourage the authors to unmask the differences observed between both set of results since the conclusions raised will be without any doubt extremely useful for the scientific community and for the future development of prevention and control of ASF. Without this information the message sent should be only circumscribed to the experimental animals used in the study, resting novelty and relevance to the study.

In fact, the conclusions raised will have to be softened.

Reviewer #2: Radulovic et al. present a detailed, well written paper describing the immune status, microbiome and transcriptional response in two groups of domestic pigs after infection with two different isolates of African swine fever virus with differing virulence. As the authors describe African swine fever is a serious threat to animal welfare and food security and the disease/immune status as well as the genetic background of animals is something that has not been widely explored. Such data will be important to help inform decisions around the potential deployment of live attenuated vaccines in the field.

The strengths of the study are the robust and detailed datasets generated, however there are a number of issues which I don't think can be realistically addressed.

1) The data is descriptive in nature, there are no experiments to demonstrate that the measured immune status/microbiome of the SPF pigs and the domestic pigs are the cause of the reported disease progression.

2) The authors results are at odds with existing data with the Estonia 2014 strain. In two separate experiments farm pigs exhibit reduced signs of disease after infection with Estonia 2014 when compared to minipigs/wild boar that were raised in an institutional facility (Zani, 2018; Sehl, 2020). One difference between these previous experiments and the author's were the challenge route (oronasal versus intramuscular).

3) Although not explored to the same depth as in the present manuscript differences between SPF and farm animals in the context of African swine fever virus infection have been previously reported (Lacasta, 2014).

Reviewer #3: The authors present results that support their hypothesis that the hygienic and thus the immunological status of a pig determines the course of the disease after ASFV infection. For this purpose, they compare SPF pigs with animals from conventional husbandry. In addition to comparative studies of naïve leucocytes and their transcriptome profiles, the disease courses and immunological reactions after infection with ASFV are described. Here, infections with highly pathogenic virus are compared with those with naturally attenuated ASF virus.

The experimental design is clearly structured. The results are presented logically and very well comprehensible according to the hypothesis and they are discussed extensively in the context of the current literature. They make a valuable contribution to the understanding of the immunopathogenesis of ASF and should be considered in particular for vaccine production based on attenuated viruses.

**Part II – Major Issues: Key Experiments Required for Acceptance**

Reviewer #1: 1) Authors should re-write the paper under the light of previous findings showing the opposite results than those here presented (not even cited).

Togehter with this main concern, authors should include additional experiments

2) Functional assays are missed (Intracellular cytokine assays, ELISPOs etc…). At least at the time of sacrifice of the surviving pigs….

3) Experiments attempting to address the mechanisms….also missed in here

Reviewer #2: (No Response)

Reviewer #3: However, I unfortunately have some criticisms concerning the gating strategy of the FACS experiments (Fig. S2 and the resulting figures).

1. The gd T cells are described as CD3+ CD4-CD8-. This is inaccurate. There is indeed a distinct population of CD8+ gd T cells with effector function, which also changes in number in the course of the infection. In the gating used here, these cells are not assigned to the gd T cells, they are found among the CD8+ ab T cells. (In addition, the gate referred to here as gd T cells also contains CD4CD8 double negative ab T cells.) If this cannot be corrected with the existing staining, it should be mentioned and evaluated in the discussion.

2. The use of the anti CD45 antibody seems incomprehensible: In Fig. S2 A and B it is used to clearly identify leukocytes. Why are CD45+ and CD45- CD3- cells used to identify NK cells in Fig. C and D? Different amounts of non-leukocytes in the individual pigs could influence the results.

**Part III – Minor Issues: Editorial and Data Presentation Modifications**

Reviewer #1: Any idea of why SPF pigs show fewer monocytes in the blood than conventional pigs? Does the same thing happen in tissues? This observation might be important since macrophages are the main ASFV target and their proportion and number might influence the infection kinetics. This issue should be better addressed in the discussion section

Reviewer #2: Some discussion of the early viremia in the Armenia infected pigs is warranted. These and related viruses have been characterized extensively using different infection routes and viremia is typically seen 3 dpi, as opposed to 1 dpi reported here. The dose used by the authors is also relatively low.

Reviewer #3: Minor points also Fig S2:

The neutophils are gated as CD45+SSChi CD14+. Consequently, they should also be named as such in the legend.

The axis labels with CD4a should only be called CD4.

PLOS authors have the option to publish the peer review history of their article (what does this mean?). If published, this will include your full peer review and any attached files.

Reviewer #1: No

Reviewer #2: No

Reviewer #3: No
---

## [Decision Letter · Decision Letter 1]

3 Jul 2022

Dear Prof. Dr. Benarafa,

We are pleased to inform you that your manuscript 'The baseline immunological and hygienic status of pigs impact disease severity of African swine fever' has been provisionally accepted for publication in PLOS Pathogens.

Best regards,

Linda Kathleen Dixon

Associate Editor

PLOS Pathogens

Klaus Früh

Section Editor

PLOS Pathogens

Kasturi Haldar

Editor-in-Chief

PLOS Pathogens

orcid.org/0000-0001-5065-158X

Michael Malim

Editor-in-Chief

PLOS Pathogens

orcid.org/0000-0002-7699-2064

Thank you for the revision to your manuscript. All reviewers comments have been answered. Reviewer 3 has a minor suggestion to change the designation in illustrations from gd T cells to CD3+ CD4- CD8- to avoid misunderstanding by readers. We agree this would improve clarity of the manuscript.

Reviewer Comments (if any, and for reference):

Reviewer's Responses to Questions

**Part I - Summary**

Reviewer #3: (No Response)

**Part II – Major Issues: Key Experiments Required for Acceptance**

Reviewer #3: (No Response)

**Part III – Minor Issues: Editorial and Data Presentation Modifications**

Reviewer #3: The authors answered my questions and took up my suggestions. The gd T cells cannot be clearly described with the staining approach used, this is now understandable mentioned in the discussion and does not detract from the results in any way. However, I would then also recommend dispensing with the designation gd T cells in the illustrations and rather designating these cells as CD3+ CD4- CD8-. Otherwise, misunderstandings could arise among readers, since using an anti gd TCR antibody would have made a clear identification possible, which is not possible here.

PLOS authors have the option to publish the peer review history of their article (what does this mean?). If published, this will include your full peer review and any attached files.

Reviewer #3: No

---

## [Editor Report · Acceptance letter]

3 Aug 2022

Dear Prof. Dr. Benarafa,

We are delighted to inform you that your manuscript, "The baseline immunological and hygienic status of pigs impact disease severity of African swine fever," has been formally accepted for publication in PLOS Pathogens.

Best regards,

Kasturi Haldar

Editor-in-Chief

PLOS Pathogens

orcid.org/0000-0001-5065-158X

Michael Malim

Editor-in-Chief

PLOS Pathogens

orcid.org/0000-0002-7699-2064